# Vegetative Propagation of the Commercial Red Seaweed *Chondracanthus chamissoi* in Peru by Secondary Attachment Disc during Indoor Cultivation

**DOI:** 10.3390/plants12101940

**Published:** 2023-05-10

**Authors:** Samuel Arbaiza, Jose Avila-Peltroche, Max Castañeda-Franco, Arturo Mires-Reyes, Orlando Advíncula, Paul Baltazar

**Affiliations:** Laboratorio de Investigación en Cultivos Marinos (LICMA), Dirección General de Investigación, Desarrollo e Innovación, Universidad Científica del Sur, Lima 15067, Peru

**Keywords:** carrageenophyte, locality, maintenance, Peru, seaweed cultivation, vegetative propagation

## Abstract

*Chondracanthus chamissoi* is an edible red seaweed with a high hydrocolloid content and food industry demand. This situation has led to a decline in their populations, especially in Peru. An alternative culture method based on the formation of secondary attachment discs (SADs) has shown several advantages over traditional spore strategies. However, there are still scarce reports of the SAD method in Peru. This work aimed to evaluate the best conditions for *C. chamissoi* maintenance prior to SAD development and the effect of locality on SAD formation using scallop shells as a substratum. Experiments were conducted with material collected from five localities in Pisco (Ica, Peru). Our results showed that the best conditions for *C. chamissoi* maintenance were: (1) fertilized seawater with Bayfolan^®^ (0.2 mL L^−1^); and (2) medium exchange every two days or weekly. These conditions reduced the biomass loss to 9.36–11.14%. Most localities showed a similar capacity to produce SADs (7–17 SADs shell^−1^). However, vegetative algae, especially Mendieta, tended to present a higher number of SADs. Vegetative fronds also showed lower levels of necrosis and deterioration compared to cystocarpic and tetrasporophytic samples. This study shows the technical feasibility of culturing *C. chamissoi* through SADs for developing repopulation and/or intensive cultivation projects in Peru.

## 1. Introduction

The edible red seaweed *Chondracanthus chamissoi* (C. Agardh) Kützing, traditionally known as “yuyo”, “mococho” or “chicoria de mar”, is distributed along the Pacific coast of South America (5–42° S) and considered endemic to the coasts of Peru and Chile [1], although there are some reports in distant regions, such as Korea, Japan, and France [2,3]. This species grows in the low-intertidal to subtidal zones (15 m depth), attached to rocky and calcareous substrates by a basal disc [4,5]. Due to its morphological variety, three forms were formerly described: f. *chauvini,* f. *lessoni*, and f. *glomeratus*. However, all of them are currently synonymized with *C. chamissoi* [3,6,7].

In Peru, *C. chamissoi* has been consumed since pre-Hispanic times and is considered a fundamental ingredient in several dishes, such as the traditional Peruvian ceviche, “picantes” (spicy), and soups [8,9,10,11,12]. However, most of the biomass (60–100% of all *C. chamissoi* exports) is destined for the extraction of carrageenans, polysaccharides with multiple applications in the formulation of various foods due to their binding, emulsifier, and thickener properties [4,13,14,15,16]. The remaining exported biomass (at most 40%) is mostly destined for China, Japan, and Taiwan, where it is traditionally used in soups and salads [17]. Thanks to its use as human food and for the hydrocolloid industry, many Peruvian coastal communities (i.e., gatherers and artisanal divers) rely on *C. chamissoi* for their economic livelihood [18,19].

In the last decades, the increasing demand for *C. chamissoi* has decreased populations, along with size reduction and lower gel quality [14]. Moreover, exports have declined during the last five years, while prices have increased sharply since 2019 [17]. This trend is also present in the internal market, where prices experienced an average increase of 366% from 2005 to 2014 [19]. In this context, cultivation efforts have increased during the last few years to achieve sustainable production of this resource [20,21]. Spore-based methods have been applied since 1998; however, they still present several limitations, such as: (1) a high mortality rate of spores during the stages of development; and (2) high operation and labor costs due to long periods of cultivation in controlled conditions (3 to 4 months) [22]. Therefore, an alternative method was proposed by Bulboa et al. [23] based on the formation of secondary attachment discs (SADs) when *C. chamissoi* thalli are anchored on a natural or artificial substratum. This new strategy for vegetative propagation presents several advantages over spore strategies: (1) lower mortality rate; (2) lower operation and labor cost due to shorter incubation time in controlled conditions (3 to 4 weeks); (3) reusability of the thalli, avoiding the massive collection of biomass from natural beds [4,23,24]. The SAD technique has been successfully applied in Chile since its first report [25,26]. In contrast, published cultivation experiences using this approach are still scarce in Peru [27]. It is worth noticing, however, that extensive use of vegetative propagation methods can lead to vigor loss, reduced production capacity, and increased susceptibility to pathogens. Thus, these methods must be combined with reproductive cell (spores or gametes)-based techniques to improve economic cultivars like *C. chamissoi* [28]. 

Factors such as reproductive phase, seasonality, substrate type, and seawater exchange have been evaluated in *C. chamissoi* vegetative cultures by SAD formation [4,23,24,25]. Biomass loss during cultivation and/or maintenance periods is critical in seaweed cultures and is attributed to excessive handling, fouling by microscopic organisms, and inadequate culture conditions [29,30,31,32]. Our research group has recorded a biomass loss of 20–35% in *C. chamissoi* prior to inoculation for SAD development [33]. However, as far as the authors are aware, suitable conditions for maintenance have not been determined for *C. chamissoi*. Another important factor is the locality (i.e., the sampling site of seaweed inoculum), which is relevant for future algal cultivators to know where to collect robust thalli to generate SADs. To the best of our knowledge, the effect of locality on SAD formation has not been assessed in this commercial red alga. Thus, this work aimed to determine the best conditions for reducing biomass loss in *C. chamissoi* fronds prior to the formation of SADs and to evaluate the effect of locality on SAD development. 

## 2. Results

### 2.1. Experiment 1: Conditions for Thalli Maintenance Prior Inoculation

BLR (%) ranged from −8.13 to 100% after four weeks in culture. Treatment with fertilized seawater, medium exchange every two days, and inoculum density of 5 g L^−1^ resulted in the lowest biomass loss (−8.13 ± 4.13%), followed by that obtained with fertilized seawater, medium exchange weekly, and inoculum density of 5 g L^−1^; 0.60 ± 14.44%) and treatment consisting of fertilized seawater, medium exchange weekly, and inoculum density of 3 g L^−1^; 1.11 ± 11.09%). Fertilization of seawater, medium exchange, and their interaction significantly affected BLR (*p* < 0.0001). The largest effect size was reported for the fertilization of seawater (ω^2^ = 0.86). Inoculum density did not affect the BLR (*p* = 0.080) (Table 1). A weekly evaluation of BLR showed that treatments using fertilized seawater did not show significant variations throughout the experiment, i.e., most of the biomass loss in these treatments was reported during the first week. A similar pattern was found in treatments using unfertilized seawater, with some exceptions where BLR increased significantly from the second week onward (Table 2). Overall, loss of biomass could be reduced to 9.36 ± 13.54% and 11.14 ± 14.27% by exchanging the medium (fertilized seawater) every two days or weekly, respectively, regardless of the inoculum density (3–7 g L^−1^) (Figure 1).

A morphological assessment of the fronds evidenced changes in all the treatments. Initially, the fragments had thalli with smooth edges and pointed pinnules and apices (Figure 2a); however, by the second week in culture, the healing process of the fronds and the growth of new shoots (1 to 2 mm in length) on the edges of the thalli and pinnules were evident (Figure 2b). By the fourth week of culture, the pinnules and apices began to show elongated and pointed shapes, narrowing and curving (Figure 2c).

### 2.2. Experiment 2: Effect of Locality on SAD Formation

All treatments showed SAD formation after 25 days of culture (Figure 3a,b). The number of SADs per shell ranged from 7 to 17. Treatment MVI (vegetative fronds from the intertidal zone in Mendieta) presented the highest formation of SADs (17 ± 2 SADs shell^−1^), followed by that obtained in treatment TVS (vegetative fronds from the subtidal zone in Talpo; 15 ± 4 SADs shell^−1^) and treatment PrVS (vegetative fronds from the subtidal zone in Punta Ripio; 15 ± 2 SADs shell^−1^). Although there was a tendency towards higher SAD values in treatments using vegetative thalli, only intertidal vegetative fronds from Mendieta (treatment MVI) showed statistically higher values (17 ± 2 SADs shell^−1^) compared to cystocarpic plants from the same zone (treatment MCI; 7 ± 1 SADs shell^−1^; *p* = 0.0052). Most of the localities showed similar values of SAD formation regardless of the reproductive state of the material and the collection zone (intertidal vs. subtidal). Only intertidal vegetative plants from Mendieta (treatment MVI) showed superior SAD formation compared to subtidal vegetative plants from Tanque Amarillo (treatment TaVS; 11 ± 1 SADs shell^−1^; *p* < 0.0001) (Figure 3a). 

The qualitative assessment showed that more than half of vegetative plants from Mendieta (treatments MVI and MVS; 54.28–62.59%) and Punta Ripio (60%; PrVS) presented the lowest level of necrosis and deterioration (<25% of the frond). Among the evaluated localities, Mendieta showed the smallest values, with 54.28–62.59% of the plants showing <25% of the frond with signs of necrosis/deterioration. Within each locality, necrosis and deterioration were usually higher in tetrasporophytic and cystocarpic plants (>25% of the frond) than in vegetative ones (<25% of the frond) (Figure 4).

## 3. Discussion

The physiological requirements of a seaweed cultivar and the laboratory protocols for its maintenance are pivotal for a successful culture [34,35]. The low availability of nutrients (especially nitrogen) limits macroalgae growth. Therefore, these elements are usually incorporated into closed cultivation systems of commercially valuable seaweeds to improve growth [36,37,38,39]. One way to incorporate nutrients is by adding agricultural fertilizers, which allow for high yields at a low cost. Various species have been cultivated by adding agricultural fertilizers: *Gracilaria chilensis* [40], *Chondracanthus squarrulosus* [41], *Porphyra* spp. [42,43], *Sarcothalia crispata* [44], *Sarcopeltis skottsbergii* (formerly *Gigartina skottsbergii*) [45,46], and *Chondracanthus chamissoi* [20,47]. Likewise, the commercial fertilizer Bayfolan^®^ (Bayer, Lima, Peru) has been widely used in macroalgae cultures, demonstrating good performance [20,43,46]. This liquid fertilizer has a concentrated nutrient formula, including vitamins and indoleacetic acid, which stimulate plant growth and could have the same effect on algae [48]. In our experiments, the use of Bayfolan^®^ with a medium exchange every two days or weekly resulted in the lowest biomass loss for *C. chamissoi* fronds, regardless of the inoculum density. In this regard, high levels of biomass loss in treatments without fertilization may be associated with a limitation of essential nutrients for seaweed metabolism [49].

Water flow has an important role in the uptake of nutrients, with low-speed flows resulting in less nutrient uptake [36,49,50]. In general, it is considered that while the flow or exchange of the culture medium is higher, the maintenance of inoculums and growth will be favored by a better uptake of nutrients [51,52,53]. For example, Bulboa et al. [23] demonstrated a greater formation and growth of SADs in *C. chamissoi* treatments with a constant flow of seawater (open flow). Similarly, Grote [54] identified the need to carry out regular water exchanges to have greater availability of nutrients for the cultivation of *Palmaria palmata*. The author pointed out that higher yields were obtained with exchange rates greater than six per day. Nevertheless, our data showed that a high frequency of medium exchange (daily) was detrimental to *C. chamissoi* maintenance. This may be due to a higher manipulation of *C. chamissoi* fragments, causing greater stress on the algae, which affected the healing process and ultimately caused biomass loss [29]. Furthermore, the physicochemical conditions of the culture medium could have been more stable in treatments with a lower frequency of medium exchange (two times a week or weekly), reducing the stress conditions. High inoculum densities can also negatively impact seaweed growth and development by reducing the availability of light, nutrients, and/or substrate [36,53,55,56,57,58]. This factor did not significantly affect the biomass loss rate in our experiments. However, we do not discard the fact that higher densities than those evaluated in this work might negatively affect *C. chamissoi* maintenance.

Besides the production of gametes and spores (tetraspores or carpospores), several studies have determined the ability of free fragments of *C. chamissoi* to re-attach to a substrate, forming basal crustose systems with the capacity to generate new fronds [59,60,61,62]. This process explains the morphological variation observed in *C. chamissoi* throughout the maintenance period, which facilitates the re-attachment by forming newly elongated, pointed, and curved pinnules and shoots [63].

Our work also demonstrated the ability of all *C. chamissoi* reproductive phases (i.e., tetrasporophyte, cystocarpic, and vegetative) from the five localities (Pisco, Peru) to propagate vegetatively by means of SAD formation on a natural substrate (*Argopecten purpuratus* shells). Similarly, Zapata-Rojas et al. [27] could induce SAD formation in two other natural substrates (clam and South Pacific abalone shells) using biomass collected in southern Peru (Moquegua region). However, the authors did not quantify the number of SAD per shell, and thus we could not compare our results with theirs. It is worth noticing that, in our work, all SADs came from vegetative propagation, as they were produced from the contact points between the inoculum (mother algae) and the substrate. No seedlings from a sporulation process were observed in any treatment. Vegetative fronds from the intertidal zone of Mendieta showed the highest SAD formation. Individuals from this locality presented suitable morphological features, i.e., two or more thick main axes and pointed apices with abundant lateral branches [47]. These characteristics would allow fronds to cover the substrate more efficiently and to receive a uniform nutrient flow. Conversely, among vegetative individuals, the ones from Tanque Amarillo showed the lowest number of SADs per shell. Their morphology, i.e., thick stems and branches (6–10 mm) with scarce secondary branches or pinnules, did not favor SAD formation. Overall, most localities showed a similar capacity to produce SADs (7–17 SADs shell^−1^). As far as the authors are aware, there are no references in the literature assessing the effect of locality on the vegetative propagation of *C. chamissoi* via SADs. According to Véliz et al. [34], the photosynthetic characteristics, pigment concentrations, antioxidant capacity, and MAA contents of *C. chamissoi* varied among populations along its distributional range along the Chilean coast, suggesting an ecotypic differentiation in this species. However, it is not clear whether this affects SAD formation in *C. chamissoi*. Localities from a wider geographical area in Peru and Chile must be included in future cultivation experiments to clarify this effect.

Regarding the reproductive phase, Bulboa et al. [23] determined that the most suitable conditions for vegetation propagation of *C. chamissoi* via SAD formation involved the use of individuals without obvious reproductive structures. This is because seaweeds allocate between 40% and 50% of the annual production of their biomass to their reproductive effort [64]. Our results confirmed this tendency only in fronds from Mendieta. Thus, locality might also be affecting the SAD formation capacity of *C. chamissoi*. In addition, the intertidal habit might affect SAD formation as individuals are naturally adapted to grow on wide rocky surfaces, forming more lateral outgrowths. Further experiments involving more localities and individuals from all reproductive phases and zones are needed to shed light on the effects of these two factors.

Necrosis and deterioration of the fronds (mainly in the apices and portions of the thallus pressed by the elastic bands) were observed in all treatments to a greater or lesser extent as a consequence of lacking an adaptation period or acclimatization. This was more critical in cystocarpic algae, as they presented irregular thallus surfaces due to the presence of reproductive structures. This feature makes the algae more prone to injuries that could become infected, generating tissue loss via necrosis. Furthermore, the irregular surface might cause “dead areas” with inadequate nutrient flow and light penetration. On the contrary, necrosis and deterioration of vegetative algae remained at acceptable levels at the end of the experiment, i.e., only 0–25% of the frond surfaces were damaged. This was more evident in intertidal vegetative fronds from Mendieta, probably because intertidal algae are more tolerant to stress than subtidal algae [65]. Some authors have found lower growth and survival rates in reproductive specimens (cystocarpic or tetrasporophytic) compared to non-reproductive (vegetative) individuals [66,67]. Furthermore, Guillemin et al. [68] reported a slower growth rate and higher mortality in reproductive fronds of *Gracilaria chilensis* compared to vegetative ones. These differences were related to lower pigment concentrations and net productivity (metabolic rates and primary productivity). 

## 4. Materials and Methods

### 4.1. Sampling Sites

*Chondracanthus chamissoi* fronds were obtained by semi-autonomous diving in subtidal natural beds (2.5–4 m in depth) at Tanque Amarillo (13°46.73′ S; 76°14.38′ W), Punta Ripio (13°47.78′ S; 76°17.65′ W), Talpo (13°48.09′ S; 76°20.74′ W), Playon (14°1.56′ S; 76°15.78′ W), and Mendieta (14°3.30′ S; 76°15.69′ W) (Figure 5). Abiotic parameters in the area were 16–19 °C, 5–6.5 mg O_2_ L^−1^, pH of 7.1–8.1, and salinity of 36.3–36.8 g kg^−1^. Fronds were transported in ice boxes at 5–10 °C to the Laboratorio de Investigación en Cultivos Marinos (LICMA) of the Universidad Científica del Sur (13°43′49.9″ S; 76°13′24.2″ W), where the experiments were carried out between March and August 2019. Specimens were cleaned as described by Macchiavello et al. [25]. 

The selected sites are located in the Ica region, on the south-central coast of Peru, a zone that is one of the most productive and commercial areas in the country, comprising more than ten *C. chamissoi* natural beds that are exploited throughout the year [69,70].

### 4.2. Experiment 1

The effect of fertilization of seawater (with Bayfolan^®^ at a final concentration of 0.2 mL L^−1^), medium exchange, and inoculum density on biomass loss rate (BLR, %) was assessed in a multifactorial experiment with three repetitions per treatment. Factor levels and conditions are shown in Table 2. Vegetative thalli (20 kg fresh weight) from Mendieta were cut into 5 ± 1 cm fragments and transferred to 1-L transparent plastic containers with seawater disinfected with sodium hypochlorite. Culture conditions were 18.5 °C, constant aeration, a 12:12-h light/dark photoperiod, and a light intensity of 30 µmol photons m^−2^ s^−1^. BLR (%) was calculated using the following formula: $$BLR\left(\%\right)=\left(\frac{W_{t_{0}}-W_{t_{n}}}{W_{t_{0}}}\right)*100$$
where $W_{t_{0}}$ is the initial fresh weight; and $W_{t_{n}}$ is the fresh weight after “*n*” weeks. BLR (%) was assessed weekly for four weeks.

### 4.3. Experiment 2

The effect of locality on the number of SADs was assessed in a unifactorial experiment (Table 3). Fronds (10 kg fresh weight) from the five abovementioned localities with 10–16 cm in length were used. In the case of Mendieta, specimens from intertidal beds were also collected. The following morphological features were considered during the selection of individuals: (a) a main axis of 4–7 mm in width with a pointed apex and thick pinnules; and/or (b) one or several erect axes with subdichotomous branches of 2–4 mm in width, abundant secondary branches and pinnules, and pointed apexes. The thalli were further separated into cystocarpic (female gametophytes), tetrasporophytic, and vegetative thalli. It is worth noticing that not all the localities presented individuals in these three reproductive phases showing suitable morphological features for SAD formation.

The cultivation method proposed by Bulboa et al. [23] was used in this experiment with some modifications. Scallop shells (8–10 cm in length) were used as a natural substratum. The shells were cleaned by submerging them in distilled water with sodium hypochlorite (0.2%) for 3 h to remove fouling and disinfect them, and then they were rinsed with distilled water. Each shell was perforated and arranged in a set of five shells with a 10 cm separation from each other. They were fixed by knots to 120 cm long ropes. This set was fixed to a PVC structure of 140 cm in length that was placed over 2.3 cm^3^ tanks containing 10 µm of filtered seawater. Two tanks were used for all treatments. *C. chamissoi* fronds were fixed over each shell by an elastic band (Figure 6). The number of experimental units inoculated (shells with fronds) was directly proportional to the amount of biomass available for each treatment. Culture conditions were 20 °C, constant aeration, under a 16:8-h light/dark photoperiod, with a light intensity of 25–30 µmol photons m^−2^ s^−1^, and a salinity of 36–37 g kg^−1^. After 25 days of cultivation, fronds were removed from the shells, and the number of SADs per shell was determined. Additionally, an arbitrary five-level scale was used to assess the necrosis and deterioration of the thalli at the end of the experiment (Figure 7). The percentage of fronds in each treatment was classified according to this scale.

### 4.4. Statistical Analyses

The effect of time (weeks) in BLR under different treatments for maintenance prior inoculation and secondary attachment disc (SAD) formation was assessed using repeated-measures ANOVA. The Bonferroni method was chosen for multiple comparisons. A three-way analysis of variance (ANOVA) was used to compare BLR under conditions of fertilization with seawater, medium exchange, and inoculum density. Normality and homoscedasticity were examined using residual plots and Levene, respectively, prior to the conduction of parametric tests. A Sidak post-hoc test was used when the results were significant. Effect size [71] was presented as ω^2^ [72] in the case of the obtaining of significant factors. The analyses were performed using the “car” [73] and “multcomp” [74] packages in R v. 4.1.2.

The number of SADs was analyzed using either a negative binomial or Poisson regression model. A likelihood ratio test was used to decide which count regression model to use. The analyses were performed using the “pscl” [75,76] and “MASS” [77] packages in R v.4.1.2. 

The significance threshold was set at *p* = 0.01 to reduce the true Type I error rate [78]. All graphs were created using GraphPad Prism 6.0 (GraphPad Software Inc., San Diego, CA, USA).

## 5. Conclusions

Fertilized seawater with Bayfolan^®^ (0.2 mL L^−1^) and a medium exchange every two days or weekly were the best conditions for the maintenance of *C. chamissoi* fronds prior to vegetative propagation, regardless of the inoculum density. SAD formation was successfully induced in fronds from all the localities, reproductive phases, and zones tested. In general, most of the localities showed a similar capacity to produce SADs (7–17 SADs shell^−1^). However, there was a tendency for vegetative algae, especially Mendieta, to present higher SADs per shell. Vegetative fronds also showed lower levels of necrosis and deterioration compared to cystocarpic and tetrasporophytic samples. This was more noticeable in vegetative intertidal fronds from Mendieta. Thus, among the localities tested, we recommend *C. chamissoi* vegetative fronds from the intertidal zone of Mendieta as the best source of biomass for vegetative propagation via SAD formation. This technique might represent a viable methodology for developing repopulation and/or intensive cultivation projects in Peru.

## Figures and Tables

**Figure 1 plants-12-01940-f001:**
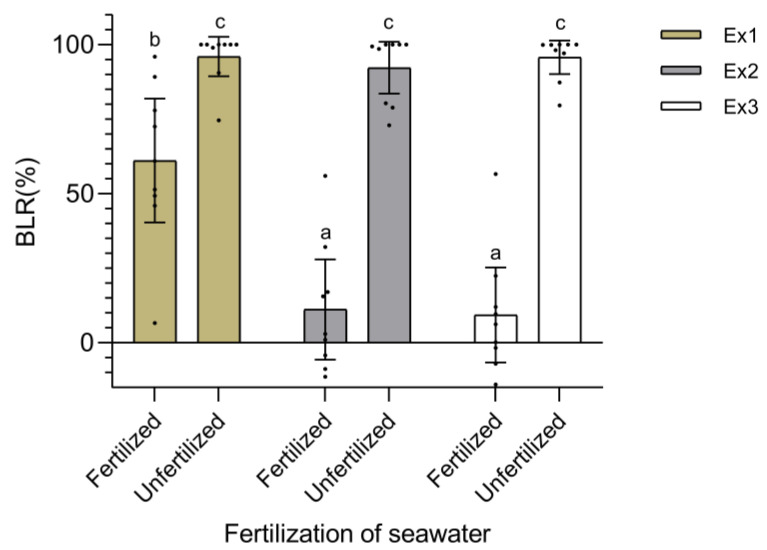
Biomass loss rate (BLR, %) of *Chondracanthus chamissoi* fronds at the end of the maintenance period (4 weeks) using fertilized and unfertilized seawater (Bayfolan^®^ at a final concentration of 0.2 mL L^−1^) under different conditions of medium exchange (Ex). Values of BLR (%) from different inoculum densities were pooled as this factor was not significant. Lowercase letters indicate significant differences (*p* < 0.01). Independent data points are shown. Error bars represent 95% confidence intervals. Ex1 = daily. Ex2 = every two days. Ex3 = weekly.

**Figure 2 plants-12-01940-f002:**
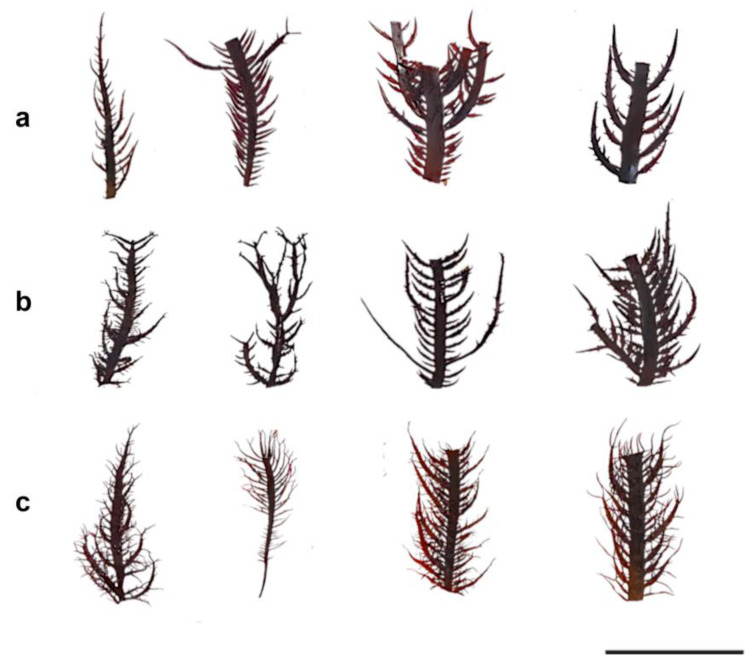
Morphological variation of *Chondracanthus chamissoi* fragments during the maintenance process in semi-controlled laboratory conditions. (**a**) Fronds at the early stage of cultivation. (**b**) Fronds after two weeks of maintenance. Note the increase in pinnules and small shoots over the entire surface of the fragment. (**c**). Fronds after four weeks of maintenance. Note the change in quantity, shape, and size of the pinnules and shoots over the entire surface of the fragment. All figures shown are on a scale of 5 cm.

**Figure 3 plants-12-01940-f003:**
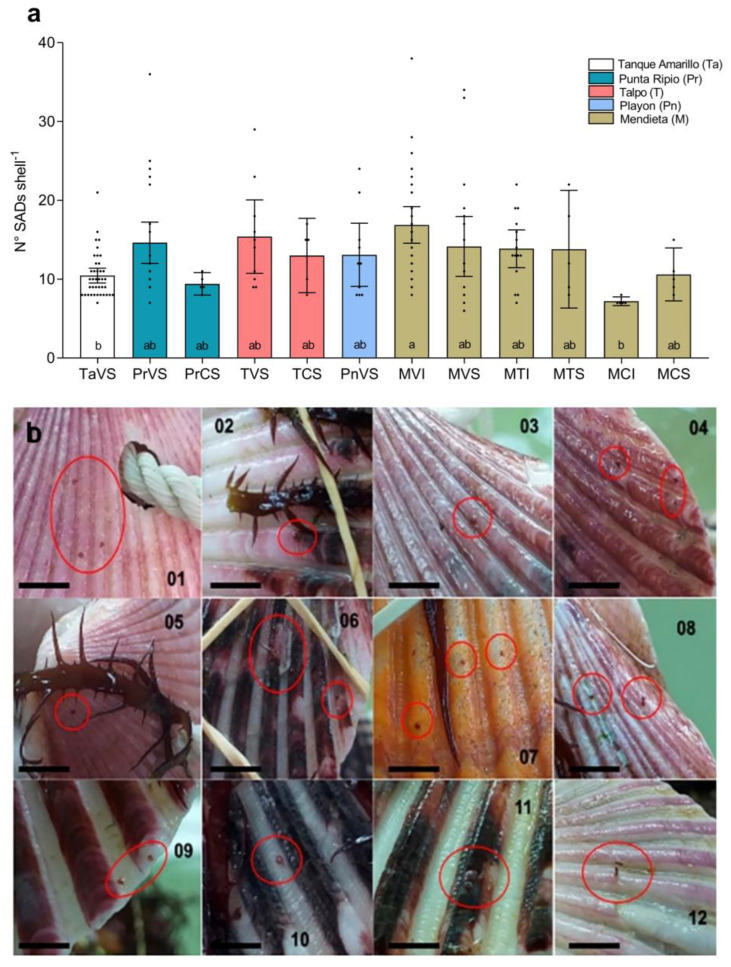
Secondary attachment disc (SAD) formation by *Chondracanthus chamissoi* in five localities. (**a**) Average number of SADs per shell. The reproductive stages considered were cystocarpic (C), tetrasporophytic (I), and vegetative (V). Most of the plants were collected from the subtidal zones (S). In the case of Mendieta, plants from the intertidal zone (I) were also used. (**b**) Representative pictures of SADs (red circles) are shown for Mendieta (01, 02, 03, 04, 05, 06), Punta Ripio (07, 08, 09), Talpo (10, 11), and Playon (12). All figures shown in (**b**) are on a scale of 1 cm. Lowercase letters in (**a**) indicate significant differences (*p* < 0.01). Independent data points are shown. Error bars represent 95% confidence intervals.

**Figure 4 plants-12-01940-f004:**
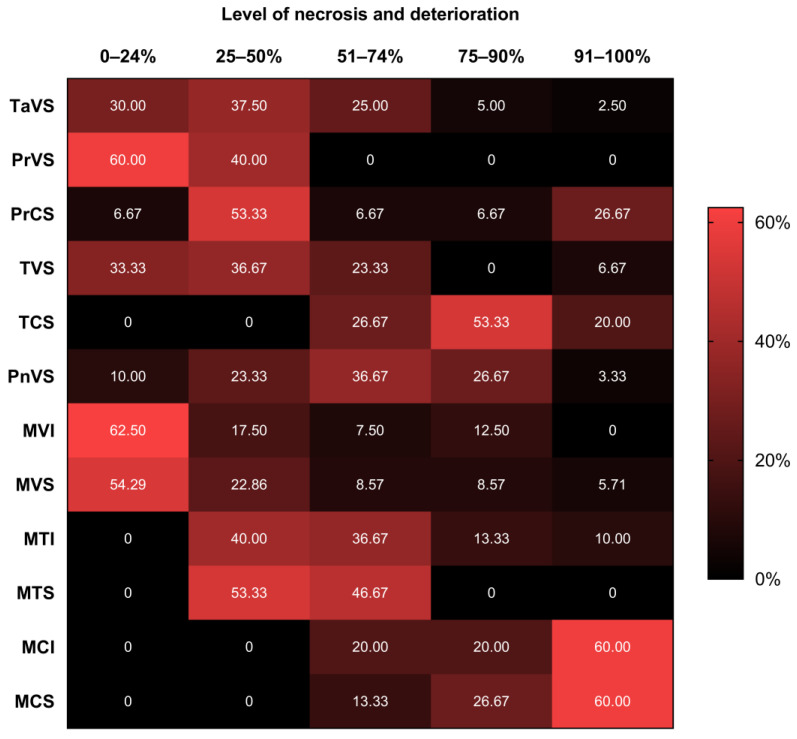
Heat map showing the percentage of fronds with different levels of necrosis and deterioration in *Chondracanthus chamissoi* at the end of secondary attachment disc (SAD) experiments (25 days). Five localities were evaluated: Tanque Amarillo (Ta), Punta Ripio (Pr), Talpo (T), Playon (Pn), and Mendieta (M). The reproductive stages considered were cystocarpic (C), tetrasporophytic (I), and vegetative (V). Most of the plants were collected from the subtidal zones (S). In the case of Mendieta, plants from the intertidal zone (I) were also used.

**Figure 5 plants-12-01940-f005:**
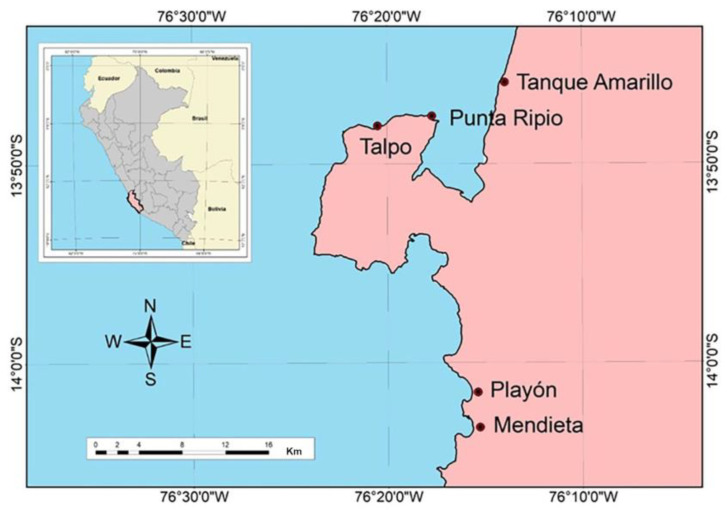
Map of sampling sites for *Chondracanthus chamissoi* (Ica, Peru).

**Figure 6 plants-12-01940-f006:**
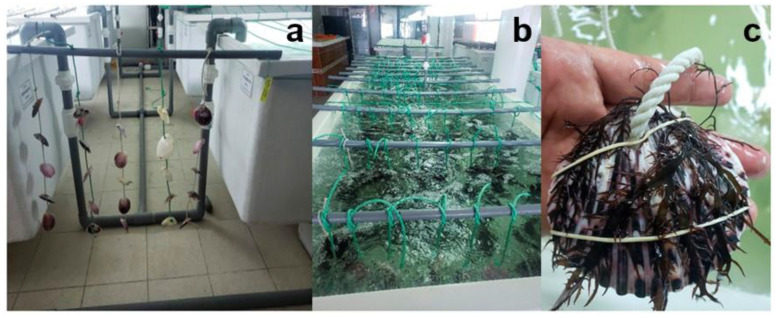
Culture system for secondary attachment disc (SAD) formation by *Chondracanthus chamissoi*. (**a**) Natural substrata (scallop shells) fixed by knots to 120 cm long ropes and anchored to a PVC structure of 140 cm in length. (**b**) Culture tanks. (**c**) *C. chamissoi* frond fixed over a shell by an elastic band.

**Figure 7 plants-12-01940-f007:**
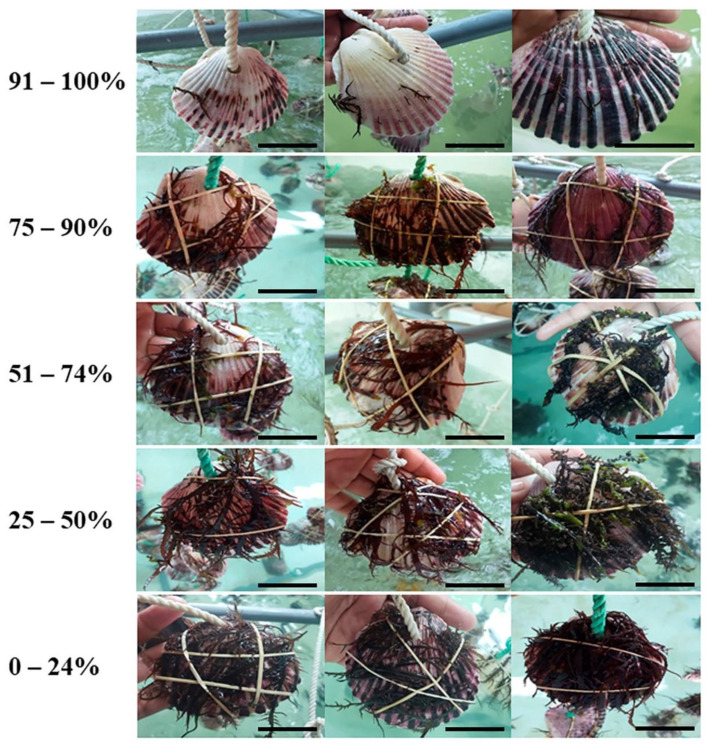
A five-level scale used for qualitative assessment of necrosis and deterioration in *Chondracanthus chamissoi* fronds at the end of secondary attachment disc (SAD) development experiments (25 days). All figures shown are on a scale of 5 cm.

**Table 1 plants-12-01940-t001:** Results of a three-way ANOVA evaluating the effect of fertilization with seawater, medium exchange, and inoculum density on the biomass loss rate of *Chondracanthus chamissoi* after four weeks in culture.

Effects	df *	*F* *	*p* *	*ω*^2^ *
Fertilization of seawater (A)	1	232.94	<0.0001	0.86
Medium exchange (B)	2	15.90	<0.0001	0.43
Inoculum density (C)	2	2.71	0.080	NS
A × B	2	13.70	<0.0001	0.39
A × C	2	1.46	0.25	NS
B × C	4	1.61	0.19	NS
A × B × C	4	1.66	0.18	NS

df * = degrees of freedom. *F* * = F statistic. *p* * = significance level. *ω*^2^ * = omega squared (effect size). NS = not significant.

**Table 2 plants-12-01940-t002:** Weekly biomass loss rate (BLR, %) of *Chondracanthus chamissoi* fronds under different treatments for maintenance prior to inoculation and secondary attachment disc (SAD) formation. Superscript letters indicate significant differences (*p* < 0.01) among weeks for each treatment. Values are presented as the mean ± 95% confidence interval (*n* = 3).

Fertilization of Seawater *	Medium Exchange **	Inoculum Density (g L^−1^)	Week 1	Week 2	Week 3	Week 4
Fertilized	Daily (Ex1)	3	3.44 ± 15.24% ^a^	14.56 ± 18.08% ^a^	54.89 ± 23.38% ^a^	74.44 ± 26.64% ^a^
		5	1.40 ± 11.30% ^a^	19.07 ± 15.32% ^a^	45.67 ± 44.57% ^a^	56.13 ± 49.44% ^a^
		7	−1.10 ± 4.25% ^a^	8.85 ± 7.54% ^a^	26.86 ± 4.20% ^a^	52.81 ± 8.59% ^a^
	Every two days (Ex2)	3	−2.89 ± 4.17% ^a^	−1.56 ± 3.30% ^a^	−5.00 ± 6.63% ^a^	7.00 ± 9.86% ^a^
		5	−4.20 ± 4.19% ^a^	−4.73 ± 2.44% ^a^	−11.80 ± 2.55% ^a^	−8.13 ± 4.12% ^a^
		7	19.71 ± 21.40% ^a^	31.90 ± 23.71% ^a^	30.71 ± 24.62% ^a^	34.57 ± 23.00% ^a^
	Weekly (Ex3)	3	−4.00 ± 8.91% ^a^	−2.44 ± 11.22% ^a^	−7.22 ± 14.98% ^a^	1.11 ± 11.09% ^a^
		5	−8.67 ± 3.40% ^a^	−1.93 ± 12.03% ^a^	−2.33 ± 10.24% ^a^	0.60 ± 14.44% ^a^
		7	21.29 ± 28.31% ^a^	27.38 ± 30.44% ^a^	23.67 ± 32.58% ^a^	26.38 ± 32.16% ^a^
Unfertilized	Daily (Ex1)	3	13.67 ± 11.51% ^a^	67.89 ± 35.67% ^a^	81.89 ± 29.50% ^a^	91.22 ± 16.23% ^a^
		5	7.40 ± 20.77% ^a^	80.47 ± 31.14% ^a^	92.00 ± 14.71% ^a^	96.87 ± 6.15% ^a^
		7	2.29 ± 9.54% ^a^	97.14 ± 1.54% ^b^	99.71 ± 0.43% ^b^	100% ^b^
	Every two days (Ex2)	3	1.44 ± 4.29% ^a^	78.89 ± 26.06% ^ab^	99.78 ± 0.43% ^b^	100% ^b^
		5	2.13 ± 4.19% ^a^	12.47 ± 7.63% ^a^	48.00 ± 22.24% ^a^	84.00 ± 14.91% ^a^
		7	3.52 ± 2.18% ^a^	38.14 ± 52.81% ^ab^	82.72 ± 33.75% ^ab^	92.76 ± 13.63% ^b^
	Weekly (Ex3)	3	10.88 ± 18.89% ^a^	59.11 ± 7.55% ^a^	70.22 ± 17.52% ^a^	88.99 ± 11.60% ^a^
		5	−2.66 ± 3.76% ^a^	95.00 ± 9.80% ^b^	98.33 ± 3.27% ^b^	99.99 ± 0.01% ^b^
		7	−6.42 ± 11.83% ^a^	93.76 ± 2.73% ^b^	96.19 ± 2.29% ^b^	98.43 ± 1.64% ^b^

* Seawater was fertilized using Bayfolan^®^ at a final concentration of 0.2 mL L^−1^. ** All the seawater was removed in each exchange.

**Table 3 plants-12-01940-t003:** Treatments used for secondary attachment disc (SAD) formation in *Chondracanthus chamissoi*.

Locality	Reproductive Stage	Zone	Treatment
Tanque Amarillo	Vegetative	Subtidal	TaVS
Punta Ripio	Vegetative	Subtidal	PrVS
	Cystocarpic	Subtidal	PrCS
Talpo	Vegetative	Subtidal	TVS
	Cystocarpic	Subtidal	TCS
Playon	Vegetative	Subtidal	PnVS
Mendieta	Vegetative	Intertidal	MVI
	Vegetative	Subtidal	MVS
	Tetrasporophytic	Intertidal	MTI
	Tetrasporophytic	Subtidal	MTS
	Cystocarpic	Intertidal	MCI
	Cystocarpic	Subtidal	MCS

## Data Availability

The data that support the findings of this study are available from the corresponding author, Paul Baltazar, upon reasonable request.

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
