# Peer review of "Vegetative Propagation of the Commercial Red Seaweed Chondracanthus chamissoi in Peru by Secondary Attachment Disc during Indoor Cultivation"

_plants, 2023, doi:10.3390/plants12101940_

Round 1
Reviewer 1 Report
The manuscript entitled "Vegetative propagation of the commercial red seaweed Chondracanthus chamissoi in Peru: optimal conditions for secondary attachment disc during indoor cultivation" addresses a very relevant and appropriate topic for this journal.
The manuscript is, on the whole, well written and well reasoned. Some taxonomy issues will have to be corrected
Corrections needed:
line 33/34 - Currently, only one form is recognized in C. chamissoi: f. glomeratus (M. Howe) S.A. Suárez [3,6]. (Note: According to international authority, the first two cited forms are currently considered synonyms of C. chamissoi)
line 79 - seawater, medium exchange every two days, and inoculum density of 5 g L-1) resulted in
line 81 - seawater, medium exchange weekly, and inoculum density of 5 g L-1; 0.60±14.44%) and
line 82 - treatment G (fertilized seawater, medium exchange weekly, and inoculum density of 3 g
line 88 - more evident when using 5 and 7 g L-1 as initial inoculum density. In contrast, when using
line 173 - ..., Sarcopeltis skottsbergii (formerly Gigartina skottsbergii) [43,44]
Reviewer 2 Report
Considering the threats to environmental stocks of seaweeds, it is very relevant the discovery of strategies to do this in an appropriate way, aiming to ensure new propagules or sustaining the cultivation from diverse alternatives. The secondary attachment disks are very relevant in this aspect, and also the finding of the best condition of keeping vegetative thalli inoculum at laboratorial controlled conditions are also peremptory.
However, some aspects of the manuscript should be considered with attention, to allow effective conclusions. These are the weaknesses of your study in my opinion:
Optimal conditions. I disagree strongly when you plan to determine an optimal condition using only a design of combining three factors’ conditions. Actually, you should use modelling experimental designs, when you conduct a range of conditions and the statistical tools allow you to determine from a model function what would be the optimal conditions considering the entire variability of levels from the relevant parameters. You fertilized your material with only one type of fertilizing… How do you know that Bayfolan is exactly the best one, if you are not comparing with other alternatives?
Another question regarding to your first experiment: why you selected the other constant conditions? For example, irradiance of 30 umol is not too low? And the temperature, is 18.5°C a good one to C. chamissoi? If I replace the cultivation to 20°C, then your data became fragile… And why to follow biomass loss? The best alternative would be to have a biomass gain and in consequence, a positive growth rates, instead of biomass losses… If your algae is losing biomass, I cannot agree that these conditions are actually optimal.
3. The comparison of localities. Actually, a place is a combination of conditions of nutrient availability, light, temperature, among other factors. So, the simple name of the place is not really informative and relevant to understand what is the meaning of using a sample from Tanque Amarillo or Mendieta. Then a characterization of the environmental abiotic data would really allow the reader to understand about if the samples came from a place with low or high nutrients, and then what could be the consequence of transferring this material to the laboratory. Moreover, to complicate, you also have different reproductive stages in the same area being compared to other reproductive stages derived from other places where the zone was different (ex. Comparison of vegetative sample from subtidal zone of Tanque Amarillo with a cystocarpic plant of intertidal area in Mendieta.
An interesting and positive aspect that you can focus more is that regardless of the place of collection, the thalli of C. chamissoi are sufficiently robust to generate SADs, what is a relevant information. A future algal cultivator can collect raw material from any of these places and he will obtain SADs that could sustain the cultivation of the species.
4. I see a disadvantage of using vegetative propagation that you should mention in your introduction at least: although other studies have success using vegetative propagules of algal thalli, I realize that if you replace entirely the spore-obtained plants with vegetative thalli you will have a lower variability (spores are formed by meiosis, and vegetative thalli came from mitosis) and you can increase the susceptibility of the samples to potential diseases or lower resistance against environmental variability. So I suggest that both strategies could be used together, not thinking in a replacement of strategy, but maybe combining both of them.
5. Think about when you insert in your discussion that more experiments are needed to allow a better conclusion of some aspect of your work… maybe these experiments should be done to complete the manuscript? L182-185 are entirely speculative and these sentences are not contributing at all to your study.
6. The figure 1 is few informative, as you have no statistical analysis to allow us to understand that the data increased or decreased significantly along the time. I suggest that you can do a one-way anova to each treatment separately, considering the factor time. Reconsider also how to show the dispersion of the data. Confidence intervals are too high, and mask the figure.
Some minor changes:
L36 – spicy in the place of “picantes”?
L65: replace “are” by “is”.
Round 2
Reviewer 2 Report
Dear authors:
I still desagree with respect to the use of the word "optimal" (including in the title!) considering the conditions for cultivation of your species. I really suggest that you change this to something different, as the conditions that you call optimal are actually only one combination of conditions, not provided by an experimental design done for this finality.
Maybe your condition is the best one considering the range that you tested but they cannot be considered as optimal.
The title can be changed to some like this: "Propagation of the commercial red seaweed Chondracanthus chamissoi in Peru by secondary attachment discs during indoor cultivation". It means that this experimental condition allowed you to cultivate and obtain the discs, but really other study focused in optimization could conclude that the optimal conditions could be another different. Then, as you did not do an optimization experiment (as RSM modelling for example), I see that this term is wrongly applied in this case.
I suggest replace the "Treatment letters" by the proper combination of conditions, then using the information found in the Table 3 (three left-columns) directly to the Table 2. In contrary, you have to mention at the Table 2 legend the meaning of each letter from A to R. Actually I see the first idea as a better choice, as your Figure 1 is presenting the conditions by their combinanion, and not as the letters A-R.
If you had more informatin about environmental conditions of each hatchery considered at your table 4, this aspect would be very relevant, as I mentined previously that "place" is an artificial denomination, actually as a result of combination of multiple abiotic parameters (temperature, wavelenghts, substrate, etc).
